# Recent Advances in Hydrogels: Ophthalmic Applications in Cell Delivery, Vitreous Substitutes, and Ocular Adhesives

**DOI:** 10.3390/biomedicines9091203

**Published:** 2021-09-12

**Authors:** Kenny T. Lin, Athena Wang, Alexandra B. Nguyen, Janaki Iyer, Simon D. Tran

**Affiliations:** McGill Craniofacial Tissue Engineering and Stem Cells Laboratory, Faculty of Dentistry, McGill University, 3640 University Street, Montreal, QC H3A 0C7, Canada; kenlin18@student.ubc.ca (K.T.L.); athw13@student.ubc.ca (A.W.); alexandra.b.nguyen@mail.mcgill.ca (A.B.N.); janaki.iyer@mail.mcgill.ca (J.I.)

**Keywords:** hydrogel, stem cells, vitreous substitutes, ocular adhesives, cataracts, retinal diseases

## Abstract

With the prevalence of eye diseases, such as cataracts, retinal degenerative diseases, and glaucoma, different treatments including lens replacement, vitrectomy, and stem cell transplantation have been developed; however, they are not without their respective shortcomings. For example, current methods to seal corneal incisions induced by cataract surgery, such as suturing and stromal hydration, are less than ideal due to the potential for surgically induced astigmatism or wound leakage. Vitrectomy performed on patients with diabetic retinopathy requires an artificial vitreous substitute, with current offerings having many shortcomings such as retinal toxicity. The use of stem cells has also been investigated in retinal degenerative diseases; however, an optimal delivery system is required for successful transplantation. The incorporation of hydrogels into ocular therapy has been a critical focus in overcoming the limitations of current treatments. Previous reviews have extensively documented the use of hydrogels in drug delivery; thus, the goal of this review is to discuss recent advances in hydrogel technology in surgical applications, including dendrimer and gelatin-based hydrogels for ocular adhesives and a variety of different polymers for vitreous substitutes, as well as recent advances in hydrogel-based retinal pigment epithelium (RPE) and retinal progenitor cell (RPC) delivery to the retina.

## 1. Introduction

According to the World Health Organization (WHO), at least 2.2 billion people suffer from some form of vision impairment, with 1 billion people suffering from moderate or severe impairments. The leading causes for these moderate or severe impairments range simply from unaddressed refractive errors to more complex age-related eye diseases [1,2,3]. In the US alone, over 4.2 million Americans above the age of 40 suffer from legal blindness or low vision due to age-related macular degeneration (AMD), cataracts, diabetic retinopathy, glaucoma, and other age-related eye diseases [4]. Due to the high prevalence of these diseases, many treatment options are available in terms of drug administration and surgery.

In terms of drug delivery, common treatments include eye drops and ointments for the anterior segment of the eye and intravitreal injections to target the posterior segment of the eye [5]. Despite these options, there exists many shortcomings with these drug delivery systems due to the extensive barriers of the eyes. Consequently, topically applied eye drops are subject to low bioavailability, leading to short dosing intervals [6]. This is explained by the high vascularization of the conjunctiva and drainage of drug formulation to the nasal mucosa (via the nasolacrimal duct) which also leads to unwanted systemic side effects [7]. Ointments are also topically applied formulations that have longer residence times at the eye due to their high viscosity [8]. However, they are more difficult for patients to apply and often cause blurred vision [9]. Intravitreal injections are the most common delivery route to treat the harder-to-access posterior segment [10]. Although this method can deliver a high drug concentration to the posterior segment, it is quite invasive and short drug retention times lead to frequent injections. Frequent injections have been shown to correlate with poor patient tolerance, endophthalmitis, retinal detachment, and vitreous hemorrhage [11,12,13]. For a review of different ocular drug delivery systems and recent advances see these comprehensive reviews [5,14].

For disorders such as cataracts, retinal detachment, and diabetic retinopathy, surgery may be necessary. Cataracts occur due to clouding of the lens, resulting in vision impairment [15]. There are different surgical methods; generally, a corneal incision is made followed by the removal of the natural lens, and the replacement of the lens with an intraocular lens (IOL) [16]. Current methods of corneal incision closure are via sutures which run the risk of tissue damage and corneal astigmatism [17,18,19,20]. Furthermore, a postoperative visit is required to remove the sutures, and the method poses risks if removal is delayed [21,22]. Surgical treatment of retinal detachment and diabetic retinopathy involve removal of the vitreous humor and replacement with gas (air, SF_6_, C_2_F_6_, C_3_F_8_) or liquids (silicone oils, perfluorocarbons) as a short-term tamponading agent [23,24,25]. However, none of these materials are long-term substitutes due to toxicity, absorption, and potential development of cataracts, glaucoma, and keratopathy [26,27]. Some substitutes also require postoperative removal. To date, there are no clinically available long-term substitutes, and this remains a challenge in ophthalmology [28]. In terms of diabetic retinopathy, it has been shown that adipose-derived stem cells (ADSC), bone marrow mesenchymal stem cells (BM-MSC), and induced pluripotent stem cell-derived products (endothelial and pericyte-like cells) can replace lost or damaged cells or provide trophic support; however, there are challenges to delivering these cells to the elusive posterior segment of the eye [29]. The routes of administration of drugs, ocular adhesives, vitreous substitutes, cells, and other treatments are illustrated in Figure 1.

Current research has looked to the application of hydrogels to overcome the challenges in drug delivery and surgical applications. Hydrogels are cross-linked hydrophilic polymers that can hold large quantities of water owing to their three-dimensional structure [32]. Their use in drug delivery has been well-documented, with many in-depth literature reviews; as such, this will not be discussed in detail [33]. In short, the use of hydrogels in drug delivery is minimally invasive with higher bioavailability, longer drug release profiles, and longer dosing intervals [30,31,34]. Instead, this review will discuss recent advances in hydrogel technology that complement surgery, such as ocular adhesives and vitreous substitutes. We will also discuss recent advances in hydrogel technology for cell and stem cell delivery to the posterior segment of the eye. Hydrogels possess many versatile characteristics that may alleviate the aforementioned shortcomings. In terms of ocular adhesives, hydrogels have the advantage of being biodegradable and bioabsorbable while having the relevant mechanical properties to serve as an effective ocular wound sealant [35]. They also show great promise as vitreous substitutes owing to their high water content, tunability, optical clarity, and similar refractive indices to the natural vitreous [36]. Lastly, hydrogels can dually serve as a delivery system and scaffold for retinal cell and stem cell delivery, as they can encapsulate cells, are permeable to nutrients, and are less invasive than solid scaffolds to install [37,38].

## 2. Hydrogel Types and Classifications

Depending on the cross-linking method employed, different types of hydrogels may be formed, each with their advantages and limitations. Physical cross-linking, which involves non-covalent bonding, does not require the use of agents that may result in adverse reactions in vivo [39]. However, the weaker bonds in physical hydrogels are prone to degradation induced by changes in conditions such as pH or temperature. While this result would be beneficial if the goal is to create a reversible hydrogel, the breakdown of hydrogel impairs its function and usually results in its dilution [40]. Conversely, chemical cro-ss-linking occurs through covalent bonding, which provides greater stability than physical hydrogels [41]. Since chemical hydrogels require the use of agents, it is crucial to employ non-toxic agents to minimize cytotoxicity and increase biocompatibility.

Hydrogels may be further differentiated by their composition of natural or synthetic polymer chains. Natural hydrogels may be protein-based, polysaccharide-based, or formed from decellularized tissues, and common biopolymers include collagen, chitosan, and hyaluronic acid [42]. Natural polymers are often components of the extracellular matrix (ECM), and, thus, they are biocompatible, biodegradable, and exhibit low cytotoxicity [42]. However, they are limited by factors such as weak mechanical properties, as compared to synthetic polymers [43]. Synthetic hydrogels are biologically inert and may be engineered to have more compatible chemical and physical properties, compared to natural hydrogels [44]. Furthermore, synthetic hydrogels allow for greater control over processes such as polymerization and degradation, which help to assess how the hydrogel should be employed in biomedical applications [44,45]. Synthetic polymers can also decrease data variability between in vitro and in vivo conditions [44]. However, even when using synthetic hydrogels alone, the structural and functional requirements to mimic the ECM are not fully met; thus, these hydrogels may be combined with natural hydrogels to form composite hydrogels to overcome these limitations [42,43].

While some hydrogels are pre-formed and administered as a gel, others operate through stimuli-responsive and in situ hydrogel systems [34]. Hydrogels may swell or de-swell in response to chemical stimuli such as pH level and ion concentration, or physical stimuli such as temperature and ultrasound [46]. Figure 2 shows how in situ hydrogels swell upon contact with a stimulus and the treatments relevant to this review that benefit from this sol-gel phase transition.

Ion-sensitive hydrogels respond to changes in the ion concentration of the environment, and an example of an ion-sensitive hydrogel is a type of poly(N-isopropylacrylamide) (PNIPAM) hydrogel derived from crown ether that was synthesized by Liu et al. (2013) [49]. Thermosensitive hydrogels may be classified as positively thermosensitive, negatively thermosensitive, or thermally reversible, and they transition to the gel phase at physiological temperature such that no additional heat source is required [34]. Positively thermosensitive hydrogels often encompass natural polymers such as gelatin, agarose, and amylose, whereas examples of negatively thermosensitive hydrogels include N-Isopropylacrylamide-based systems and polysaccharides such as methyl cellulose [50]. pH-sensitive hydrogels swell and undergo gel formation at physiological pH due to the ionization of functional groups found on the polymer chains [51]. Examples of natural pH-sensitive polymers include alginic acid (alginate), hyaluronic acid, and chitosan [34], and examples of synthetic pH-sensitive polymers include poly(L-glutamic acid) (PGA), poly(histidine) (PHIS), and poly(aspartic acid) (PASA) [52]. Ultrasound-responsive hydrogels are activated upon receiving ultrasonic energy, and an example of an ultrasound-responsive hydrogel system was demonstrated by Kubota et al. (2019), who fabricated tungsten particle-capsulating calcium alginate microbeads to release drugs upon the application of ultrasound [53]. Hydrogels may also have near infrared (NIR) light-responsive properties, which have been explored in areas such as photothermal therapy (PTT) and chemotherapy. Wang et al. (2021) incorporated gold nanorods (AuNRs) into an antibacterial hydrogel (CP@Au@DC_AC50) to treat uveal melanoma (UM), a type of malignant intraocular tumor [54]. Another example of NIR light-responsive hydrogel is composed of β-glycerophosphate-bound chitosan (CGP), dopamine-modified alginate (Alg-DA), and AuNRs, and was developed by Zeng et al. (2019) [55]. Lee et al. (2021) introduced an alternative method of forming NIR light-triggered hydrogels from molybdenum disulfide (MoS2) nanoassemblies and thiol-functionalized thermo-responsive polymers [56]. Another type of responsive hydrogel is the magnetism-responsive hydrogels, which respond to an external magnetic field [57]. For instance, Zhang et al. (2019) have incorporated iron oxide (Fe_3_O_4_) nanoparticles into the tetra-PEG/agar hydrogel network to introduce magnetism-responsive properties in tissue engineering for the treatment of injured tissues [58]. In situ forming systems offer several advantages, such as prolonged contact time with the site of drug absorption, improved patient compliance and level of comfort, and minimized precorneal elimination of the drug, which are beneficial for advances in ocular use [34,59].

## 3. Hydrogels Used in Cell and Stem Cell Delivery

Retinal degenerative diseases (RD) such as retinitis pigmentosa (RP), diabetic retinopathy, glaucoma, and age-related macular degeneration (AMD) can cause great negative impacts to one’s eyesight. These diseases usually result in the destruction of photoreceptors or retinal ganglion cells (RGCs) [60,61]. A stem-cell approach to replace the damaged neurons has been shown as a viable method towards dealing with RD [62]. The advantage of this type of therapy is that stem cell differentiation is thought to replace the damaged cell population. One of the major challenges is the survival and engraftment of transplanted stem cells. It has been shown that the injection of a suspension of retinal progenitor cells (RPC) only results in a fraction of the cells surviving [63]. In addition, it has been shown that survival of transplanted cells increases when delivered with a biodegradable scaffold [61]. The ideal scaffold should be biocompatible, biodegradable, and bioabsorbable and should direct cell adhesion, differentiation, and proliferation [64]. Traditionally, solid scaffolds based on synthetic polymers such as poly(L-lactic acid)/poly(lactic-co-glycolic acid) (PLLA/PLGA), poly(3-caprolactone) (PCL), and poly(glycerol-sebacate) (PGS) have been used [61]. These scaffolds are inflexible and implantation of these structures in the subretinal space is invasive and may cause retinal detachment [65,66]. Recently, much research has focused on using injectable hydrogels as a delivery and scaffold system for stem cells [67]. Hydrogels have the advantage of being high in water content, capable of encapsulating cells, being similar in structure to the ECM, being permeable to nutrients, and being less invasive than solid scaffolds to install [37,38]. This section will report on recent advances in hydrogel technology in the delivery of cells and stem cells.

In an in vitro study by Kim et al. (2019), the authors looked at the feasibility of polyethylene glycol (PEG)/Gellan Gum (GG) hydrogel as a scaffold and delivery system for retinal pigment epithelium (RPE) cells [64]. The PEG/GG gel was a physical mixture, and FT-IR confirmed no cross-linking reactions between the two polymer types. They studied the PEG/GG gel at four different concentrations of PEG: 0, 1, 3, and 5 wt% with GG having a fixed concentration of 1%w/v. They found that, with increasing PEG concentration, pore size and porosity decreased while viscosity increased. Biodegradation of the gel also seemed to depend on PEG concentration with initial degradation rates decreasing as PEG content increased. Live/Dead assays and MTT analysis showed that 3 wt% PEG content was optimal due to the comparably high RPE cell survival and proliferation rates. In addition, expression of relevant genes in RPE cells (RPE 65, CRALBP, and NPRA) [68,69,70] was the highest in the 3 wt% PEG/GG compared to all other concentrations. This gel has the potential to be a promising scaffold in tissue engineering; however, this study did not directly test the biocompatibility of the PEG/GG. Further in vitro tests must be performed to adequately assess cytotoxicity to then warrant investigations in vivo to further establish the efficacy of this hydrogel. A ternary hydrogel of gelatin (Ge)/gellan gum (GG)/glycol chitosan (CS) was also investigated for RPE delivery [38]. FT-IR analysis showed that this hydrogel was a physical mixture and that there were electrostatic interactions between anionic GG and cationic CS. Comparisons were made between Ge/GG and Ge/GG/CS hydrogels, and Rim et al. (2020) found that the inclusion of CS slowed down gel degradation and increased compressive strength. Live/Dead staining results of RPE cells after 28 days and dsDNA content showed that cell proliferation rates were higher in the Ge/GG/CS gel compared to the Ge/GG gel. Morphological characterization and histological studies showed that the Ge/GG/CS gel provided an optimal microenvironment for encapsulated cells for cell-cell and cell-matrix interactions. Gene expression analysis of encapsulated RPE cells also showed greater expression of RPE pertinent genes (RPE65, CRALBP, MITF, NPR-A, RHODO, COL I) [68,69,70,71,72] in GE/GG/CS hydrogels, which are indicative of higher amounts of proliferative cells. Despite the promising results, in vivo studies must be conducted to fully determine its biocompatibility and efficacy. Gellan gum grafted with dopamine (DFG) was also evaluated as an RPE delivery system [73]. Morphologically, DFG gels were highly porous and contained smaller pores than GG gels. The reduced pore sizes may enhance cell proliferation due to increased specific surface area for cell attachment [74,75]. The mass swelling ratio was higher in DFG gels, owing to the increased degree of hydrophilicity presented by the dopamine groups. This may be beneficial, as enhanced swelling properties lead to the absorption of culture media needed for cell function [76]. DFG gels also showed lower gelation viscosity, temperature, and rate, which was attributed to the bulky nature of dopamine loosening the GG’s helix structure [77]. These changes in gelation properties allowed for lower injection forces, conferring a superior injectability of DGF gels. In terms of cell viability, DFG gels had less cell apoptosis events possibly due to a favorable microenvironment created by the catechol moiety [78]. Just as in the study by Rim et al. (2020), more work in vivo is needed to fully determine biocompatibility and efficacy.

Jiang et al. (2019) synthesized an injectable hydrogel that is made up of natural polymers and is capable of self-healing for use in stem cell delivery to the retina [79]. The hydrogel (CS-Odex) is a crosslinked polymer of chitosan hydrochloride (CS) and oxidized dextran (Odex) via Schiff-base linkages. Gelation time could be tuned by the concentration of CS, with an increase in CS leading to a decrease in gelation time. Pore sizes of the hydrogel were observed to be adequate in size for nutrient transportation and cell metabolism (120 μm to 180 μm). Similar mechanical properties to the retinal soft tissue and favorable degradation profiles (tunable by CS concentration) point to CS-Odex as a suitable candidate for RPC delivery. In terms of self-healing properties, the gel was able to maintain its viscoelastic properties (measured via G′ and G″ modulus) at low strains of <70% and recover its original viscoelastic properties back at low strains after exposure to damage-inducing 100% strains. In addition, the hydrogel was shown to merge both macroscopically and microscopically after being split in half. Live/Dead assay and inflammatory and apoptotic factor expression analysis showed hydrogel-cultured RPC cells to be cytocompatible and that the self-healing properties conferred a higher survival rate post-injection (~90%) compared to the control (~84%). Biocompatibility was shown via in vivo studies on nude rats. CS-Odex was also shown to enhance cell-proliferation rates due to the stimulation of Akt and Erk pathways (pathways involved in proliferation) [80,81], with increasing CS inducing further stimulation. They also found that RPC cells preferentially differentiated into retinal neurons, with higher CS increasing preferential differentiation. Overall, the self-healing properties, high proliferation rates, and preferential differentiation into retinal neurons presents the CS-Odex hydrogel as a promising RPC delivery platform to treat those with RD. In a similarly goaled study, Park et al. (2019) investigated the potential for an in situ cross-linking hydrogel to be a vehicle for retinal stem cell delivery [66]. Specifically, they tested gelatin-hydroxyphenyl propionic acid (Gtn-HPA), which is a biodegradable polymer that can first be injected and then undergo gelation in vivo. They employed horseradish peroxidase and peroxide to assess in vitro compatibility and in vivo graft survival of a mixture of Gtn-HPA conjugate and RPC suspension upon enzyme-mediated gelation. In terms of cell survival and proliferation, they found that the Gtn-HPA hydrogel system was compatible with the RPCs, while demonstrating minimal apoptosis. Based on the anti-leukocyte staining, Gtn-HPA-delivered grafts were observed to exhibit a decrease in inflammatory response. Furthermore, in vivo results showed that there were more eyes with surviving cells in the gel-cell mixture cohort, relative to the saline-delivered control. Thus, they concluded that this hydrogel may increase the chances of cell survival upon transplantation, and that it is promising as a vehicle for retinal stem cell delivery; however, future work should study the effects of incorporating growth factors into the hydrogel and should continue to investigate more gelatin-based cross-linking hydrogels to help advance in situ retinal tissue engineering. In another study, Tang et al. (2019) also studied the effects of hydrogels derived from gelatin-hyaluronic acid (gel-HA) on RPC behavior, which included cell survival, proliferation, and differentiation [82]. They formed gel-HA hydrogels with and without mussel-inspired polydopamine (PDA) and observed that the hybrid hydrogels (i.e., gel-HA-PDA) offered decent biocompatibility to support processes such as cell adhesion, survival, and delivery. Furthermore, they found that gel-HA-PDA hydrogel improved neuronal differentiation and cell migration and adhesion, which could be attributed to the strong adhesive property of these hydrogels. They also found that the gel-HA hydrogel promoted cell proliferation, thus demonstrating its potential for RPC proliferation in transplantation therapy. These results provide a deeper understanding of the development and usage of biomaterials for RPC-based transplantation therapy. Dormel et al. (2020) also utilized gelatin-based hydrogels and hypothesized that replacing phosphate buffered saline (PBS) with gelatin-based hydrogel as the cell carrier might minimize the effects of shear stress on cells as they are injected through a small-bore needle [83]. Thus, they injected PBS as the cell carrier versus injecting a gelatin-based hydrogel, gelatin-hydroxyphenyl propionic acid (Gtn-HPA), through a 31-gauge needle, and analyzed their effects on cell viability and proliferation, as well as the phenotypic expression of human RPCs (hRPCs). They found that hRPCs in the PBS group that had experienced shear stress had a reduction in cell viability by 50% or more as evidenced by increased cell apoptosis and decreased cell proliferation; whereas hRPCs in the hydrogel group did not experience shear-induced change in cell viability nor proliferation, thus leading them to conclude that biomaterial hydrogels may be a suitable cell carrier replacement for PBS in RPC therapy. Table 1 provides a summary of the discussed hydrogels for RPE and RPC delivery.

## 4. Hydrogels Used as Vitreous Substitutes

Diabetic retinopathy is a severe vitreoretinal disease that leads to vision loss because of damage to retinal blood vessels and neurons. Damage to these structures is due to abnormally high levels of blood sugar [23,24]. Surgical intervention usually involves removal of the natural vitreous and replacement with a vitreous substitute, but comes with potential risks as the substitute may act as a scaffold for proliferative vitreoretinopathy (PVR) or diabetic membrane, resulting in recurrent retinopathy [23,24,25,84]. The natural vitreous is a transparent gel-like structure that is in the space between the lens and the retina. It is composed of 98% water, hyaluronic acid, and different types of collagen [26]. The current challenge is developing a vitreous substitute that fulfills all the requirements of replacing the natural vitreous: (1) clear and transparent; (2) inert; (3) similar refractive index and density to the natural vitreous; (4) similar viscoelastic properties to the natural vitreous; (5) sufficient mechanical rigidity; (6) non-absorbable and non-biodegradable; (7) hydrophilic; (8) maintains normal IOP; (9) injectable through small-gauge needles; (10) allows for circulation of ions and electrolytes; (11) easy to manipulate; (12) self-renewable to require a single implantation [30,85]. Current substitutes include different types of gases and liquids [26]. Air is an inexpensive, colorless, and nontoxic vitreous substitute, but has limited tamponade capabilities as it is rapidly absorbed by the blood [86]. Sulfur hexafluoride (SF_6_) and perfluorocarbon (C_3_F_8_) are more common options as they can persist for longer periods of time (6–8 weeks). Nevertheless, gases are expandable, and patients must maintain a constant altitude to avoid fluctuations in IOP [87]. Perfluorocarbon liquids are clear and colorless and have been used as temporary tamponade agents during surgery; however, failure to remove these substances results in retinal toxicity and intraocular inflammation [88]. Silicon oils are great tamponading agents owing to their surface tension, but are prone to emulsification and long-term persistence may result in cataracts, corneal toxicity, or glaucoma [27].

Hydrogels have shown promise in meeting the aforementioned criteria owing to their high water content, tunability, optical clarity, and similar refractive indices to the natural vitreous [36]. Current efforts have focused on in situ forming hydrogels, which can be injected as a liquid and gels upon a stimulus change. This is because hydrogels would otherwise irreversibly shear upon injection, causing destruction of the hydrogel network and breaking up crosslinks [26]. Many studies in the past few years have failed to create a hydrogel, both natural and synthetic, without any shortcomings; thus, there are no hydrogels in clinical use as vitreous substitutes to our knowledge [28]. This section will focus on recent advances in promising hydrogels.

A feasibility study by Jiang et al. (2018) looked at the physical and rheological properties of HPCTS-ADA hydrogel [89]. HPCTS-ADA hydrogel was prepared via a self-crosslinking reaction between hydroxypropyl chitosan (HPCTS) and alginate dialdehyde (ADA) via Schiff base formation. This hydrogel had similar properties to the natural vitreous in terms of water content, pH, density, refractive index, and optical transmittance. Cytotoxicity tests in vitro demonstrated cytocompatibility and in vivo tests on rabbits (slit-lamp observation, intraocular pressure, corneal endothelium examination, B-scan ultrasound, and fundus photography) showed no significant adverse events such as inflammation, opacity, or abnormal IOPs. An electroretinogram and histopathologic examinations showed a decrease in retinal function, indicating that it was not completely biocompatible. Another study by Wang et al. (2021) also looked at chitosan-based hydrogels [90]. They studied an in situ hydrogel of cross-linked polymer (CMCTS-OHA) of oxidized hyaluronic acid (OHA) and carboxymethyl chitosan (CMCTS). Mechanistically, crosslinking occurred via the covalent attachment of the OHA aldehyde groups to the CMCTS amino groups via a Schiff base reaction, at physiological conditions. Water content and refractive indices and densities of these gels were very similar to those of the natural vitreous. The hydrogel had high compressive strength, being able to endure mechanical stress, which could present as an excellent shock absorber to external forces on the eye. Transmittance of the hydrogel was similar to that of the natural vitreous, with higher cross-linked hydrogels exhibiting less transmittance. CMCTS-OHA also demonstrated self-healing and biodegradable properties. In vivo and in vitro tests showed this to be a great candidate as an ideal vitreous substitute, as no damage to the retina nor any toxic reactions occurred even after a 90-day assessment.

Other gels with self-healing properties also include PanaceaGel SPG-178, which is a self-assembling peptide gel [91]. Its self-assembling properties are due to its ability to spontaneously self-assemble into nanofibers and create stable β-sheets. At 1% (wt/vol), the hydrogel had a similar refractive index to the natural vitreous and could transmit visible light (transmission rate 96.7%). The gel was shown to be biocompatible and no damage to the retinal tissue, diseases, or cataracts were found. The self-assembling properties of the gel allowed for self-healing when damaged via injection through a small-gauge needle. This allowed hydrogel that had been broken down by the shearing forces of injection to reform into a gel state once in the vitreous cavity. Further testing is needed to properly evaluate this gel in terms of longer-term biocompatibility studies. Its ability to act as a tamponading agent also needs to be evaluated along with the ability to remove the vitreous substitute once installed. Nevertheless, these findings widen future possibilities when it comes to using smaller gauge systems.

A two-component hydrogel of thiolated gellan and poly(methacrylamide-co-methac -rylate-co-bis(methylacryloyl-cystamine)) (poly(MAM-co-MAA-co-BMAC) yields a the- rmoresponsive hydrogel that is aqueous at 45 °C and gels at body temperature [92]. Cross-linking occurred via thiol oxidation. The hydrogel was shown to swell and increase pressure, evidencing its efficacy as a potential tamponading agent. The degree of swell depended on the concentration of poly(MAM-co-MAA-co-BMAC) utilized in the hydrogel solution. The gel was also shown to have similar refractive indices and density compared to the natural vitreous with a transmittance of more than 83% to visible light. In vitro toxicity studies showed that biocompatibility and in vivo preclinical studies on rabbits showed no signs of inflammation nor cataracts, and transparent corneas after a four month evaluation; however, partial opacities were viewed via a portable slit-lamp. These results are promising, and future work is focused on running clinical trials. On the topic of thermoresponsive gels, Xue et al. (2020) looked at the factors that modulated transparency of a poly[(R)-3-hydroxybutyrate-(R)-3-hydroxyhexanoate] (PHBHx)-based thermogel [47]. Specifically, the hydrogel is a copolymer of poly(PHBHx/PEG/PPG urethane) (PHxEP), with the PEG group being hydrophilic, the PPG group being thermosensitive, and the PHBHx group being hydrophobic. They measured the optical transparency of PHxEP gels at different PHBHx concentrations and found that, at low (0.5%wt) concentrations, PHxEP had >90% transmittance at 37 °C to 500–600 nm wavelengths. Increasing PHBHx concentrations (2, 5, 8%wt) resulted in cloudy gels with transmittances <5%. In regard to the low PHBHx concentration, the gel remained in the sol phase at room temperature (25 °C) and gelled at physiological temperature, with sufficient mechanical strength to qualify as a vitreous substitute. The gel was biocompatible, displaying no signs of inflammation, having normal IOP, and retaining retinal structure over a 180-day period. Xue et al. (2020) posited that excessive hydrophobic interactions in thermogels can lead to aggregations that lead to opacities, explaining why low PHBHx concentrations resulted in clear gels. They extrapolated that hydrogels with shortcomings in transparency can be tuned by adjusting the level of hydrophobic interactions via the polymer composition. A similarly-designed thermoresponsive hydrogel, EPC, was synthesized from hydrophilic PEG, thermosensitive PPG, and hydrophobic and biodegradable poly(ε-caprolactone) (PCL) linked via urethane bonds [93]. EPC at 7% concentration was deemed to be optimal, as too low, or too high, of a concentration resulted in retinal and cytotoxicity. EPC-7% was also optically clear and had a similar refractive index to the natural vitreous. It has also been shown to act as a long-term tamponading agent, able to last up to one year. A major point of interest of EPC hydrogel is its regenerative properties, demonstrated in vivo. EPC-7% biodegrades in about three months; however, it promotes the eventual formation of a vitreous-like body. Proteomic analysis found that this vitreous-like body was similar in composition to the natural vitreous (924 out of 1177 natural vitreous proteins identified). Reformation of the vitreous-like body was a pleasant surprise, as the traditional consensus was that the vitreous is unable to reform [94]. It is unknown how this occurs, but it is likely that the mechanical properties of the hydrogels somehow impact tissue regeneration via mechano-sensing mechanisms [95,96]. This finding precipitates a novel approach to designing hydrogels as vitreous substitutes. Current and past research has focused on synthesizing non-biodegradable hydrogels with prolonged degradation rates for long-term use. This research paves the way for hydrogels that may be biodegradable and transient, but also promotes the regeneration of the natural vitreous [97].

Baker et al. (2021) studied the physical and chemical properties of HA-oxime hydrogels [98]. These hydrogels were composed of hyaluronan modified with aldehyde (HAA) or ketone (HKA) cross-linked to PEG-tetraoxyamine (PEGOA₄). They found the gelation rate to be easily tunable, as the higher the ratio of HAA relative to HKA, the faster the gelation rate. Density and refractive indices were similar to the natural vitreous and the hydrogel remained transparent even during degradation. They also showed that the hydrogel did not undergo swelling, as it degrades and maintained normal IOP ranges (13.50–25.92 mmHg) in the rabbit models. Evaluation of in vivo stability showed that the HA-oxime hydrogel has a half-life of 43 days and completely degrades after 300 days, acting as a sufficiently long tamponading agent for after retinal detachment surgery. In vitro and in vivo testing showed cytocompatibility and biocompatibility with no signs of redness, inflammation, and corneal or vitreous haze. The HA-oxime hydrogel showed much promise when compared to similarly composed hydrogels being studied. Examples include a hydrazone-crosslinked HA hydrogel commencing clinical trials (Vitagrus ABV-1701) that has been shown to swell and cause increased IOP in ~30% of patients [99]. Another example is Healaflow^®^ (Anteis S.A., Plan Les Ouates, Switzerland), which is a hydrogel formed by cross-linking of hyaluronic acid to 1.4-Butanediol diglycidyl ether (BDDE) [100]. It is commercially available, and FDA approved as a space filler in glaucoma surgery. Despite being biocompatible, it has a very short retention time and can only serve as a short-term tamponading agent. Much of the success of Baker et al. (2021)’s HA-oxime hydrogel is attributed to the click cross-linking system that was employed. The click chemistry cross-linking system allows for hydrolytically stable bonds and obviates the need for any catalysts, cross-linking agents, or UV activation which may cause harm to the eye [101].

Many of the aforementioned hydrogels focused on making a hydrogel with physical characteristics that are as similar to the natural vitreous as possible; however, mimicking its chemical functionalities is just as important. Vitamin C in the natural vitreous facilitates a steep oxygen gradient where oxygen concentration is low near the lens epithelial cells and high near the retinal pigmented epithelial cells [102]. Vitrectomy depletes this vitamin C and the oxygen gradient dissipates, resulting in a higher oxygen concentration near the lens [103]. This is thought to promote the formation of cataracts and may be why there is such a high incidence of cataracts after vitrectomy [104]. A study by Tram et al. (2020) looked at poly(ethylene glycol) methacrylate (PEGMA) and poly(ethylene glycol) diacrylate (PEGDA)-based hydrogels [105]. Specifically, they loaded vitamin C onto PEGDA and PEGDA-co-PEGMA hydrogels and determined the feasibility of these gels as both a vitreous substitute and in maintaining the oxygen gradient. They found that both gels had similar viscoelastic properties to the natural vitreous, were resistant to shearing (capable of injections through small 22- or 33- gauge needles), and were 90% transparent to visible light (similar to the natural vitreous). In vitro testing also showed cytocompatibility. Refractive indices for the PEGDA and PEGDA-co-PEGMA gels were 1.3350 and 1.3359, further demonstrating the viable optical properties (natural vitreous is 1.3349). The release of vitamin C from the two gels was shown to protect against oxidative damage from reactive oxygen species. Surprisingly, they found that using an unloaded hydrogel also had a protective effect against oxidative damage. Together, this demonstrated a synergistic effect of the hydrogel and vitamin C in reducing reactive oxygen species activity; however, the mechanism for how the hydrogel confers this protection needs to be further studied. More research is also needed in designing a hydrogel with the ideal vitamin C release rates and retention times to better protect the lens from oxidative damage, as current results show complete vitamin C degradation in five days. In a follow-up study on the effects of vitamin C in the vitreous chamber, they found that glutathione was able to significantly extend the vitamin C stability, with 70% remaining after 14 days [106]. They also found that, although physiological concentrations (1–2 mM) of vitamin C were cytotoxic in vitro, adding glutathione increased cell viability back up to 90–100%. Glutathione is also known to recycle lens-damaging oxidized vitamin C [107]. Future related studies should investigate optimizing a hydrogel with vitamin C and glutathione to decrease the prevalence of post-operative cataracts. Table 2 provides a brief summary of the discussed hydrogels investigated as vitreous substitutes.

## 5. Hydrogels Used as Ocular Adhesives in Corneal Wounds and Incisions

There is a high prevalence of ocular injuries, with 3% of all visits to the emergency room being attributed to eye trauma [33]. Most of these injuries are at the level of the cornea and can become vision threatening. Along with corneal incisions performed during cataract surgery, corneal injuries can simply self-heal; however, severe corneal injuries most commonly require suturing or adhesives [108]. As mentioned earlier, suturing comes with its various drawbacks, including astigmatism, tissue damage, and endophthalmitis [17,18,19,20,21]. Suturing also requires a skillful surgeon and is not the ideal wound closure method when compared to other wound closure alternatives. One alternative is ocular adhesives, which are designed to solve the drawbacks of sutures. Historically, cyanoacrylate-based adhesives (non-FDA approved) have been used off-labelled by ophthalmologists; however, studies have demonstrated various cytotoxicities with this type of adhesive [109]. Even though these adhesives are quick and easy to administer, they can rapidly degrade into accumulating cyanoacetate and formaldehyde, inducing inflammation [110]. Another popular adhesive is fibrin, which is a blood-based material that has been shown to form a smoother seal, thus providing greater patient comfort. However, it has risks associated with transmitted diseases from blood donors, and fibrin glue prepared from the patient’s own blood is expensive, cannot be processed instantaneously, and yields variable concentrations of the product [111].

Hydrogel-based adhesives have been developed to contribute to the solution of sutureless surgery. Development of a hydrogel-based adhesive requires the following design considerations in addition to biocompatibility and biodegradability: (1) adhesion to wet corneal surfaces; (2) controlled and effective polymerization or gelation to close the corneal wound; (3) restore intraocular pressure; (4) solute diffusion properties for corneal healing; (5) sufficient elasticity; (6) bio-absorbed or removed as corneal tissue regenerates [112]. According to Oelkler and Grinstaff (2008), numerical requirements for an ideal ophthalmic adhesive include leak pressure (>80 mmHg), cross-linking time (<30 s), mechanical properties (5–200 kPa), swelling (<200%), diffusion coefficient (>2107 cm^2^ s⁻^1^), refractive index (1.32–1.40), cytotoxicity (passed based on ISO standard), adhesion strength (>0.1 kPa), viscosity (5–100 cP), degradation time (1 week–6 months), and resident time on wound (1 day–6 months) [112]. Fitting these standards, hydrogel adhesives have been developed from a variety of polymers. These include polyethylene glycol (PEG) [113], dextran [114], dendrimers [115], chondroitin-sulfate [116], hyaluronic acid, collagen, and gelatin [35]. Functionalization of these polymers and research into using multiple platforms as co-polymers has led to the development of many potential hydrogel adhesives. This section will present recent advances in hydrogel ocular adhesives. For a comprehensive review of current and past ocular adhesives, refer to Trujillo-de Santiago et al. (2019) [35].

To date, there is only one hydrogel-based adhesive that has been approved by the FDA: ReSure^®^ Sealant developed by Ocular Therapeutix. This adhesive is polyethylene glycol (PEG)-based and is principally used in sealing clear corneal incisions. The sealant’s flexible nature allows it to conform easily to wound structures and to form a continuous barrier that covers deepithelized surfaces until healing has completed [117]. In a study by Masket et al. (2014), they evaluated the degree of fluid egress in ReSure^®^ hydrogel sealant versus sutured cataract incisions (CCI) [108]. They found that 97.6% of incisions leaked when not sealed at all, supporting the evidence that self-healing corneal incisions are prone to wound leakage [118,119]. In terms of hydrogel sealant versus sutures, they found that 4.1% of hydrogel-sealed incisions leaked versus the 34.1% of sutured incisions after the first week of surgery. They also found that there were less minor adverse events (subconjunctival hemorrhaging, eye irritation, and foreign-body sensation) as compared to the suture group. Another study by Nallasamy et al. (2017) compared surgical characteristics of applying the ReSure^®^ sealant versus not applying the sealant using a 1:1 matched cohort of exposure-discordant eyes [120]. The corneal incisions were cases where sutures would otherwise not have been needed with no wound leaks one day postoperatively. They found no significant difference in surgical time, intraocular pressure, corneal edema, or foreign body sensation when comparing sealant versus non-sealant eyes. This showed that applying the sealant does not diminish surgical efficiency and supports the use of ReSure^®^, where installation of intraocular lenses (IOLs) (multifocal, accommodating, toric) that require precise and stable positioning is critical. More recent studies have validated the resistance of ReSure^®^ to increases in IOP when applied to 3 mm clear corneal incisions. Shehata et al. (2021) demonstrated the aforementioned using an ex vivo rabbit eye model and burst pressure tests [121]. Fredell and Hamill (2019) also demonstrated usage of the sealant outside CCIs in cataract surgery, but also in Descemet membrane endothelial keratoplasty (DMEK) [122]. They found that sealant-closed wounds were able to withstand the high IOP that occurs due to postoperative graft adherence. The sealant also contributed a water-tight seal without any instances of wound reopening or leakage.

There has also been headway with other polymeric platforms that have made their way into a clinical setting. OcuSeal™, developed by Beaver-Visitec International, is a dendrimer-based hydrogel bandage used to stabilize ocular wounds such as corneal incisions [123]. Although it is not FDA approved, it is CE (European Conformity) marked. In a clinical trial by Uy and Kenyon (2013), they compared surgical induced astigmatism (SIA), foreign body sensation, and wound edge closure rates across three different wound closure groups [124]. The groups consisted of stromal hydration (control), suture, and the ocular bandage. They found that the ocular bandage group had less instances of SIA compared to the suture group, the ocular bandage group had the least foreign-body sensation compared to the other two groups, and the ocular bandage and suture group had a greater proportion of successful wound closure compared to the control. Work done by Kenyon, Qiao, and Lee (2014) has found that the bandage can create a barrier that blocks penetration of *Staphylococcus aureus* and *Pseudomonas aeruginosa* for at least 24 h, indicating its usefulness in preventing endophthalmitis [125]. Disadvantages of OcuSeal™ include a cure time that is too rapid, making it difficult for proper application, and that it has only been shown to seal CCIs of 2.75 mm [124].

Dendrimer-based polymers have also been combined with other platforms to yield successful adhesives. Work being done by Kambhampati et al. (2020) has yielded hydrogel sealants based on dendrimer-hyaluronic acid [126]. This OcuPair^TM^ sealant consisted of methacrylated hydroxyl dendrimer (D-MA) and methacrylated hyaluronic acid (HA-MA). Exposure to blue light initiated the cross-linking. At the right ratios, OcuPair^TM^ is transparent, flexible, and can withstand IOPs of over 70 mmHg. The sealant can adhere to the cornea for up to five days and appears to be biocompatible to the cornea. Kambhampati et al. (2020) states that this sealant has the potential to treat warzone-sustained corneal injuries as a temporary corneal wound stabilizer. Further in vitro, in vivo, and clinical tests are needed to further evaluate this potential along with a method to mass synthesize this hydrogel.

Gelatin-based hydrogel adhesives are natural-based polymers that have the advantages of biocompatibility and biodegradability [34]. Gelatin is derived from collagen, which is endogenous to the corneal stroma and sclera [127]. GelCORE is a hydrogel adhesive that is biocompatible, cytocompatible, and effective in terms of sealing corneal defects and promoting re-epithelialization [128]. The adhesive is gelatin-based and uses a visible light cross-linking system via free radical polymerization, in the presence of type 2 initiator Eosin Y, co-initiator triethanolamine (TEA), and co-monomer N-vinylcaprolactam (VC) [129,130]. The visible light photocrosslinking system is FDA-approved [131] and has advantages over UV light cross-linking systems, which may cause retinal damage [132], corneal sunburns [132], and carcinogenesis [133]. The mechanical properties (compressive and elastic modulus) have been shown to be tunable via changes to GelCORE concentration and cross-linking time. The enzymatic degradation rate has been shown to be tunable via in vitro testing. At 20% GelCORE concentration, the adhesives showed greater adhesion and cohesion capabilities as compared to PEG-based (CoSEAL) and fibrin-based (Evicel) adhesives. In vitro and in vivo tests showed cytocompatibility and the facilitation of corneal tissue regeneration. Further studies have evaluated the potential of GelCORE to dually serve as a drug-eluting bioadhesive. Khalil et al. (2020) devised a method to load micelles containing ciprofloxacin (CPX) within the hydrogel for infection control and inflammation suppression [134]. To load GelCORE with CPX-loaded micelles, gelatin methacryloyl (GelMA), the aforementioned cross-linking solution, and CPX-loaded micelles were combined and vortexed at 37 °C to form the prepolymer solution. Cross-linking was performed via exposure to visible light. Burst pressures of GelCORE loaded with the micelles (GelCORE + MC) showed no significant difference when compared to the unloaded GelCORE (both around 35 kPa). Wound closure tests showed that adhesive strength and properties also remained the same in both GelCORE and GelCORE + MC, regardless of the incision size and substrate used in the ex vivo testing. In terms of antimicrobial properties, in vitro and ex vivo testing showed GelCORE + MC to be effective against Gram-positive and Gram-negative bacteria. Further testing also showed biocompatibility, cytocompatibility, and cytoprotection to corneal cells against infection. Overall, the GelCORE drug delivery system has the potential to be very targeted, with longer drug residence times and lower dosage requirements; however, in vivo testing is needed. Other gelatin-based hydrogels have been investigated for their use as an ocular adhesive. Gelatin-glycidyl-methacrylate hydrogel (GELGYM) is a derivative of GelMA synthesized from gelatin solution with glycidyl methacrylate in PBS [135]. The advantage of this method over GelMA is that GELGYM can potentially hold double the methacrylate groups as each amine bearing amino acid can covalently bond to two glycidyl methacrylates. This abundance of methacrylate groups enables photocrosslinking at very low intensities (20 mW/cm^2^) using the same initiator compounds (eosin Y, TEA, VC) as in the GelCORE preparations [128]. It has also been shown that GELGYM has tunable mechanical properties based on the degree of methacrylation (FD), cross-linking time (CT), and GELGYM concentration. GELGYM has been shown to have 5- to 10- fold higher enhancement of mechanical properties when compared to GelMA, owing to greater FD and crosslinking density. UV-Vis spectroscopy also showed that GELGYM has similar transparency to the cornea, dependent on CT. GELGYM was biocompatible and ex vivo data showed that GELGYM can seal full penetrating corneal defects of up to 4 mm in diameter, compared to 2 mm-diameter defects via GelMA. GELGYM may serve as a potential adhesive outside ophthalmology, being shown to adhere to various biological surfaces such as the aorta, heart, and muscle, to name a few, with greater adhesive strength than most widely used adhesives in these areas. Table 3 provides a brief summary of the discussed hydrogels investigated as ocular adhesives.

## 6. Conclusions

Recent advances in hydrogel technology have been instrumental in the development of hydrogel-based cell and stem cell delivery systems, vitreous substitutes, and ocular adhesives, thus highlighting its importance in the treatment of ocular diseases. While many studies have demonstrated the great potential of hydrogel-based treatments, future work should continue to investigate in vivo and run clinical trials, in order to further apply this valuable technology in clinical settings. A careful and holistic evaluation of the literature has guided us in forming unique perspectives of hydrogels and their applications in these three areas of interest. For RPE and RPC delivery, we believe that future research should continue looking into hydrogels with high self-healing capabilities, such as the ones developed by Jiang et al. (2019), and facile injectability [79]. These factors will lead to easy administration and higher post-injection cell survival rates. Many other studies have performed the necessary in vitro, ex vivo, and in vivo tests; thus, we look forward to any clinical testing that may arise. In terms of vitreous substitutes, we believe that there is much work to do before hydrogels become a clinical option; however, continued focus into hydrogel research will undoubtedly yield promising results. This is highlighted by the reformation of a vitreous-like body as demonstrated in Liu et al. (2019)’s work, which will shift how researchers approach hydrogel design for vitreous substitutes [93]. Additionally, hydrogel design for vitreous substitutes will also need to focus on mimicking chemical functionalities of the natural vitreous as demonstrated by Tram et al. (2020) who studied the role of vitamin C in mimicking the oxygen gradient of the natural vitreous [105]. Lastly, hydrogels as ocular adhesives are already in clinical use, and we believe that current investigations into gelatin-based hydrogels will lead to further optimization of existing options. As hydrogels can be drug-loaded, there is also the possibility that ocular adhesives dually serve as drug-eluting mechanisms to manage any inflammation or infections that may accompany an incision or wound. To put it succinctly, the various studies showcased in this review have highlighted the enormous potential of hydrogel-based treatments for ocular diseases and their impactful role in the future.

## Figures and Tables

**Figure 1 biomedicines-09-01203-f001:**
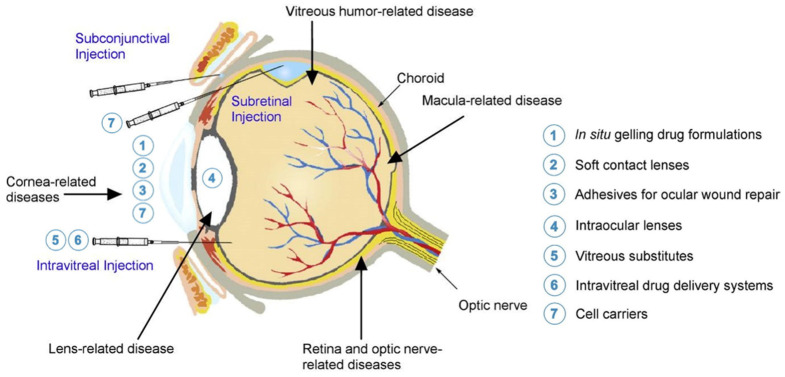
Anatomy of the eye labelled with location of ocular diseases. Numbers correspond to the type of treatment and show the targeted ocular structure or the route of administration. Reprinted with permission from ref. [30]. Copyright 2015 Elsevier and Reprinted with permission from ref. [31]. Copyright 2017 Elsevier.

**Figure 2 biomedicines-09-01203-f002:**
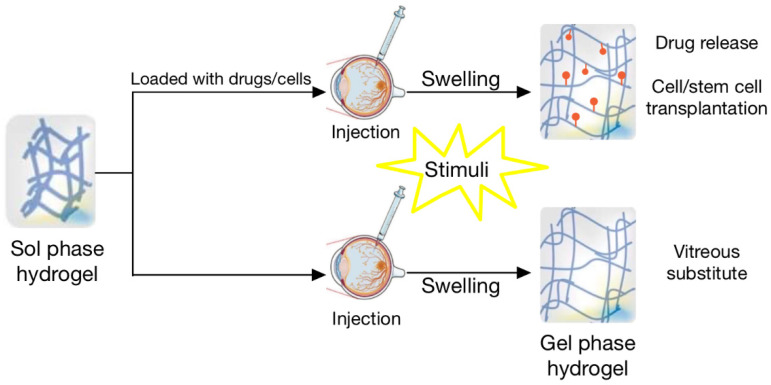
Sol phase hydrogel is liquid and injectable. A chemical or physical change after administration to the eye acts as a stimulus that causes the transition to the gel phase. The hydrogel swells and can serve as a drug-eluting mechanism, cellular scaffold, or vitreous substitute. Reprinted with permission from ref. [47]. Copyright 2020 Royal Society of Chemistry and Reprinted with permission from ref. [48]. Copyright 2015 Creative Commons Corporation (this figure is a derivative of “Figure 2” by Fathi et al. (2015) used under CC BY-NC 4.0) [48].

**Table 1 biomedicines-09-01203-t001:** Summary of hydrogels investigated for stem cell delivery and scaffolding.

Type of Hydrogel	Polymer(s)	Retinal Cell Type	In Vitro Cytocompatibility	In Vivo Biocompatibility	Notes
PEG/GG [64]	Polyethylene glycol and gellan gum	RPE	Yes—Live/Dead assay, MTT analysis, and gene expression analysis	Not conducted	None
Ge/GG/Cs [38]	Gelatin, gellan gum, and glycol chitosan	RPE	Yes—Live/Dead staining and gene expression analysis	Not conducted	Inclusion of Cs lead to higher proliferation rates
DFG [73]	Gellan gum grafted with dopamine	RPE	Yes—Live/Dead staining and gene expression analysis	Not conducted	Dopamine confers superior hydrogel injectability and a favorable microenvironment
CS-Odex [79]	Chitosan hydrochloride and oxidized dextran	RPC	Yes—Live/Dead assay and inflammatory and apoptotic factor expression analysis	Yes—H&E staining and Masson’s trichrome staining	Hydrogel is able to self-heal. CS increases preferential differentiation towards retinal neurons.
Gtn-HPA [66]	Gelatin-hydroxyphenyl propionic acid	RPC	Yes—Live/Dead assay	Yes—immunohistochemistry and anti-leukocyte staining	Gtn-HPA lowers proliferative potential and transplants show persistent retinal detachment
gel-HA [82]	Thiolated gelatin and methacrylated hyaluronic acid	RPC	Yes—Live/dead staining, inflammatory and apoptotic factor expression levels, and cell adhesion analysis	Not conducted	Improved cell proliferation
gel-HA-PDA [82]	Thiolated gelatin, methacrylated hyaluronic acid, and polydopamine	RPC	Yes—Live/dead staining, inflammatory and apoptotic factor expression levels, and cell adhesion analysis	Yes—H&E staining and Masson’s trichrome staining	Improved neuronal differentiation and cell migration and adhesion

**Table 2 biomedicines-09-01203-t002:** Summary of hydrogels investigated as vitreous substitutes.

Type of Hydrogel	Polymers	In Vitro Cytocompatibility	In Vivo Biocompatibility	Notes
HPCTS-ADA [89]	Hydroxypropyl chitosan and alginate dialdehyde	Yes—MTT assay	No—electroretinogram and histopathologic anaylsis	None
CMCTS-OHA [90]	Oxidized hyaluronic acid and carboxymethyl chitosan	Yes—MTT assay	Yes—H&E staining	Self-healing properties
PanaceaGel SPG-178 [91]	13 amino acid peptide (RLDLRLALRLDLR)	Yes—Live/Dead staining	Yes—slit lamp examination, fundoscopy, electroretinography, histopathology	Self-assembling properties prevent damage from injection
poly(MAM-co-MAA-co-BMAC [92]	Thiolated gellan and poly(methacrylamide-co-methac -rylate-co-bis(methylacryloyl-cystamine))	Yes—ECIS and CellTiter-Glo Luminescent Cell Viability end-point assay	Yes—electroretinography, optical coherence tomography, and H&E staining	Thermoresponsive
PHxEP [47]	Poly[(R)-3-hydroxybutyrate-(R)-3-hydroxyhexanoate], polyethylene glycol, and polypropylene glycol	Yes—MTT assay	Yes—histopathological examination	Thermoresponsive
EPC [93]	Poly(ε-caprolactone), polyethylene glycol, and polypropylene glycol	Not conducted	Yes—slit-lamp examinations, fundus evaluation, electroretinography, H&E staining	Degrades and regenerates a vitreous-like body
HA-oxime [98]	Hyaluronan modified with aldehyde or ketone and PEG-tetraoxyamine	Yes—Live/Dead assay	Yes—H&E examination	Click chemistry cross-linking system obviates the need for cross-linking agents
PEGDA [105]	Poly(ethylene glycol) diacrylate	Yes—CellTiter-Glo luminescent cell viability assay	Not conducted	Can be loaded with vitamin C to protect against oxidative damage
PEGDA-co-PEGMA [105]	Poly(ethylene glycol) methacrylate and poly(ethylene glycol) diacrylate	Yes—CellTiter-Glo luminescent cell viability assay	Not conducted	Can be loaded with vitamin C to protect against oxidative damage

**Table 3 biomedicines-09-01203-t003:** Summary of hydrogels investigated as ocular adhesives.

Type of Hydrogel	Polymer	Corneal Incision Length (mm)	Burst Pressure (kPa)
ReSure^®^ Sealant [121]	Polyethylene glycol	<3.5	12.4
OcuSeal™ [136]	Poly(glycerol succinic acid) and PEG-aldehyde	<2.8	26.4
OcuPair^TM^ [126]	Methacrylated hydroxyl dendrimer and methacrylated hyaluronic acid	<6	9.3
GelCORE [128]	Methacrylated gelatin	<3	30.1
GELGYM [135]	Glicydlmethacrylated Gelatin	<4	26.7

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
