# Peer review of "Recent Advances in Hydrogels: Ophthalmic Applications in Cell Delivery, Vitreous Substitutes, and Ocular Adhesives"

_biomedicines, 2021, doi:10.3390/biomedicines9091203_

Round 1

Reviewer 1 Report

In this manuscript, recent advances in hydrogel technology that have been instrumental as significant numbers of hydrogel-based cell and stem cell delivery systems, vitreous substitutes, and ocular adhesives in the treatment of ocular diseases are well described comprehensively based on the significant expertise of the authors. This manuscript would be informative for various readers including ophthalmologists and researchers engaging in both basic and translational research. However, there is still some room for improvement. The reviewer strongly recommends the authors address the comments below to improve this manuscript. 

Major comment:

Please prepare a concise summary table of substrates that are mentioned in each section from section 2 and later including the development stage, i.e., in vitro study only, in vivo study completed, and so on. It is easier for readers to look back and compare each substrate for better understanding due to the volume of information in this manuscript. It would be some amount of work for the authors, however, it is a great help for readers to understand the contents of this manuscript better.

Minor comments

  1. raw 58-61: Replacement with gas or liquids, such as silicone oil, perfluorocarbons are used as a short-term tamponading agent, not a replacement of vitreous, please rewrite to clarify this point to prevent misunderstanding from readers.
  2. raw 66-67: iPS cells cannot be directly used for replacement therapy, but iPS cell-derived cell products will do so. Please revise.
  3. raw 322-323: as one of the potential risks of artificial vitreous replacement, it could act as a scaffold for PVR or diabetic membrane, leading to additional traction on the retina and recurrent retinopathy (Foster et al, Expert Rev Ophthalmol, 2008). Please mention this point as well. 

Reviewer 2 Report

The manuscript by Lin et al. represents a review article discussing discuss recent advances of hydrogel technology in surgical applications, including dendrimer and gelatin-based hydrogels for ocular adhesives and a variety of different polymers for vitreous substitutes, as well as recent advances in hydrogel-based retinal pigment epithelium and retinal progenitor cell delivery to the retina. This manuscript is timely, is well written and easy to follow. However, I have several suggestions on how to improve the presentation:

- The authors have excluded from the review several topics connected to hydrogels (for example hydrogels in drug delivery), while the title of the review is still very broad. I would like to suggest authors thinking about how to make it more focused on the topics covered by the text;

- The manuscript has a great potential from the point of view of illustrations. In particular, in my opinion, the chapter ‘Hydrogel Types and Classifications’ should be illustrated by the scheme or a table helping reader’s understanding the content;

- I think it is very important to know authors’ opinion on the most promising directions of further research. These ideas could be included into 'Conclusions' section, which is extremely short within the current version. 

Reviewer 3 Report

This review summarizes recent advances of hydrogels in ophthalmic applications. I have several concerns:

  1. There are no figures in this manuscript, I am confused about this. I suggest authors to provide a representative figure/scheme in each section.
  2. Authors should give more descriptions on the advantages of hydrogels in ophthalmic applications in INTRODUCTION.
  3. Are there any other responsive hydrogels? For instance, NIR laser-responsive or magnetism-responsive hydrogels. If yes, please give more discussion in INTRODUCTION.
  4. Authors should show their own perspectives on future development of hydrogels in ophthalmic applications.
  5. I suggest authors to list all the abbreviations at the end of the manuscript.

Round 2

Reviewer 3 Report

I'm satisfied with the revisions. This manuscript can be published.